# Effects of Pirfenidone and Nintedanib on Markers of Systemic Oxidative Stress and Inflammation in Patients with Idiopathic Pulmonary Fibrosis: A Preliminary Report

**DOI:** 10.3390/antiox9111064

**Published:** 2020-10-30

**Authors:** Alessandro G. Fois, Elisabetta Sotgiu, Valentina Scano, Silvia Negri, Sabrina Mellino, Elisabetta Zinellu, Pietro Pirina, Gianfranco Pintus, Ciriaco Carru, Arduino A. Mangoni, Angelo Zinellu

**Affiliations:** 1Department of Medical, Surgical and Experimental Sciences, University of Sassari, 07100 Sassari, Italy; 30048416@studenti.uniss.it (V.S.); silvia.negri86@gmail.com (S.N.); pirina@uniss.it (P.P.); 2Department of Biomedical Sciences, University of Sassari, 07100 Sassari, Italy; esotgiu@uniss.it (E.S.); sabrinamellino3@gmail.com (S.M.); gpintus@uniss.it (G.P.); carru@uniss.it (C.C.); azinellu@uniss.it (A.Z.); 3Unit of Respiratory Diseases, University Hospital Sassari (AOU), 07100 Sassari, Italy; elisabetta.zinellu@aousassari.it; 4Department of Medical Laboratory Sciences, College of Health Sciences and Sharjah Institute for Medical Research, University of Sharjah, Sharjah P.O Box 27272, UAE; 5Department of Clinical Pharmacology, College of Medicine and Public Health, Flinders University and Flinders Medical Centre, Adelaide 5100, Australia; arduino.mangoni@flinders.edu.au

**Keywords:** idiopathic pulmonary fibrosis, inflammation, nintedanib, oxidative stress, pirfenidone

## Abstract

Introduction: In vitro evidence suggests that pirfenidone and nintedanib, approved agents for the treatment of idiopathic pulmonary fibrosis (IPF), exert anti-inflammatory and anti-oxidant effects. We aimed to investigate such effects in vivo in IPF patients. Methods: Systemic circulating markers of oxidative stress [nuclear factor erythroid 2–related factor 2 (Nrf2), thiobarbituric acid- reactive substances (TBARS), homocysteine (Hcy), cysteine (Cys), asymmetric dimethylarginine (ADMA) and ADMA/Arginine ratio, glutathione (GSH), plasma protein –SH (PSH), and taurine (Tau)] and inflammation [Kynurenine (Kyn), Tryptophan (Trp) and Kyn/Trp ratio] were measured at baseline and after 24-week treatment in 18 IPF patients (10 treated with pirfenidone and 8 with nintedanib) and in 18 age- and sex-matched healthy controls. Results: Compared to controls, IPF patients had significantly lower concentrations of reduced blood GSH (457 ± 73 µmol/L vs 880 ± 212 µmol/L, *p* < 0.001) and plasma PSH (4.24 ± 0.95 µmol/g prot vs 5.28 ± 1.35 µmol/g prot, *p* = 0.012). Pirfenidone treatment significantly decreased the Kyn/Trp ratio (0.030 ± 0.011 baseline vs 0.025 ± 0.010 post-treatment, *p* = 0.048) whilst nintedanib treatment significantly increased blood GSH (486 ± 70 μmol/L vs 723 ± 194 μmol/L, *p* = 0.006) and reduced ADMA concentrations (0.501 ± 0.094 vs. 0.468 ± 0.071 μmol/L, *p* = 0.024). Conclusion: pirfenidone and nintedanib exert beneficial effects on specific markers of oxidative stress and inflammation in IPF patients.

## 1. Introduction

Idiopathic pulmonary fibrosis (IPF), a progressive disorder of unknown etiology, is characterized by interstitial lung fibrosis which leads to respiratory failure and premature death [1,2]. Although there is significant interindividual variability in disease progression (e.g., from clinically stable to rapidly progressive), the life expectancy without treatment is two to three years [3]. IPF is linked to several interacting genetic and environmental risk factors, which cause a recurrent damage of alveolar epithelial cells. This, in turn, triggers an aberrant repair process leading to chronic inflammation and, ultimately, fibrosis [4,5,6]. Although the excessive inflammatory response in IPF is likely multifactorial, the transforming growth factor-β (TGF-β) signaling pathway plays a significant role [7]. Oxidative stress (OS) is also a key player in the pathogenesis of IPF [8,9]. While the lungs are particularly sensitive to OS, due to their exposure to relatively elevated oxygen tensions, pollutants and exogenous oxidants further promote OS stimulating inflammatory cells to produce reactive oxygen species (ROS) and free radicals, such as hydrogen peroxide, hydroxyl radical, and superoxide radical. OS might further enhance the inflammatory state through the activation of nuclear factor- kB (NF-kB), with consequent activation and recruitment of immune cells [10]. Albeit the role of inflammation and OS as causative factors of IPF is controversial, both processes are up-regulated throughout the disease progress [11,12,13].

In 2014, the Food and Drug Administration approved nintedanib and pirfenidone for IPF treatment because of their ability to retard disease progression [14,15]. Pirfenidone is an antifibrotic drug with well-known anti-inflammatory and antioxidant properties [16,17]. Nintedanib is a tyrosine kinase inhibitor with anti-angiogenic effects through the blockade of the vascular endothelial growth factor (VEGF) pathway [18]. More recently, studies have also reported that nintedanib exerts anti-oxidant and anti-inflammatory effects [19,20]. However, the anti-inflammatory and anti-oxidant effects of pirfenidone and nintedanib have been primarily observed in in vitro studies [16,17,18,19,20]. Therefore, we sought to determine the presence of such effects in vivo by investigating a comprehensive panel of markers of OS and inflammation in IPF patients before and after 24-week treatment with pirfenidone or nintedanib and in a group of age- and sex-matched healthy participants. In particular we assessed the plasma concentrations of: (i) Nrf2, a transcription factor that controls the expression of enzymes and proteins responsible for cellular defence against OS; (ii) TBARS, a group of aldehydic substances (principally malondialdehyde) deriving from lipid peroxidation and the most frequently used biomarker of oxidative stress; (iii) GSH, a tripeptide thiol found at high concentrations in cells where it exerts antioxidant effects, reducing ROS such as superoxide or the hydroxyl radical; iv) PSH, the total protein –SH groups that act as a key redox buffer in plasma, scavenging reactive oxygen and nitrogen species; (v) homocysteine and cysteine, low-molecular mass thiols involved in free radicals and hydrogen peroxide production due to the high reactivity of their thiol group; (vi) ADMA and arginine, amino acids commonly found in human circulation, involved in nitric oxide (NO) production. Oxidative stress may lead to a shift in the ADMA/arginine ratio, toward ADMA accumulation, resulting in diminished NO synthetase (NOS) activity and subsequently lower NO availability; (vii) tryptophan and its degradation product kynurenine, well recognized biomarkers of immune activation. Systemic inflammation is commonly associated with increased kynurenine and/or decreased tryptophan concentrations.

## 2. Methods

### 2.1. Patients

A consecutive series of eighteen newly diagnosed IPF patients were recruited at the Respiratory Unit of the University of Sassari between 2018–2019. All patients signed a written consent and the study was approved by the ethics committee of the University Hospital of the University of Cagliari (PG/2018/4426). IPF was diagnosed according to the current evidence-based guidelines for the diagnosis and management of IPF [21]. High-resolution computed tomography (HRCT) images and lung-biopsy samples of individual patients were reviewed by two experienced radiologists and two experienced pathologists, respectively. Each diagnosis was confirmed during multidisciplinary meetings attended by local respiratory, pathology, and radiology experts in interstitial lung disease. Patients with the following conditions were excluded from the study: currently in a period of acute exacerbation of IPF; comorbid conditions including malignancy, bleeding tendency, and severe hepatic or renal dysfunction; use of immunosuppressants, interferon, D-penicillamine, and colchicine, or oral corticosteroids during the preceding three months. Ten patients were treated with pirfenidone 2403 mg/day and eight patients with nintedanib 300 mg/day according to the ATS ERS IPF treatment guideline [22]. Completion of a 24-week treatment period was followed by a follow-up visit 2 weeks later. Spirometry tests were performed in accordance with the criteria published by the American Thoracic Society and the European Respiratory Society [23] at baseline and after 24 weeks of treatment. A group of 18 healthy controls matched for age, sex and smoking status, with no medical history, was also included.

### 2.2. Biochemical Analysis

Blood was collected in EDTA tubes after an overnight fast and centrifuged at 4 °C and 1500× *g* for 10 min to separate plasma, which was stored at −80 °C until analysis.

#### 2.2.1. ADMA and Arginine

Arginine and ADMA, were assessed by capillary electrophoresis UV detection as previously described [24]. Since it has been reported that ADMA analysis assay imprecision may increase the chance of statistical type 2 errors in clinical studies and lead to a significant strength underestimation of the association between ADMA and other biochemical or clinical variables [25], we used a capillary electrophoresis method that provide inter-assay CV between 2 and 3% for arginine and ADMA measurement. A volume of 100 µL of plasma were mixed with 50 μL (100 μmol/L) of internal standard homoarginine; 300 μL of acetonitrile/ammonia (90/10) were then added to precipitate proteins. After centrifugation at 3000× *g* for 5 min, the clear supernatant was evaporated under a vacuum centrifuge and the residue was re-dissolved with 200 μL and injected in capillary electrophoresis. A MDQ capillary electrophoresis system equipped with a diode array detector was used (Beckman Instruments, CA, USA). The analysis was performed in an uncoated fused-silica capillary, 75 μm ID and 60.2 cm length (50 cm to the detection window), injecting 10 mm of water plug followed by electrokinetic injection of 10 kV for 200 s and a buffer plug of 4.5 mm. Separation of analytes was carried out in a 50 mmol/L Tris buffer titrated with 1 mol/L phosphoric acid to the pH 2.30, 15 °C, at normal polarity and at 12 kV. Arginine and ADMA were detected at 200 nm.

#### 2.2.2. Kynurenine and Tryptophan

Kynurenine and tryptophan were measured by capillary electrophoresis UV detection as previously described [26]. A total of 100 μL of plasma was mixed with 50 μL of methyltryptophan as internal standard (50 μmol/L) and 1 mL of cold acetonitrile. After centrifugation, 1 mL of supernatant was evaporated under vacuum and the residue was redissolved with 80 μL of water. The sample was then injected in capillary electrophoresis. A MDQ CE system equipped with a diode array detector was used (Beckman Instruments, CA, USA). The analysis was performed in an uncoated fused-silica capillary, 75 µm ID and 30 cm length (20 cm to the detection window), injecting 90 nL of sample. The separation was carried out in a 100 mmol/L Bis-Tris propane buffer titrated with phosphoric acid (1 mol/L) to pH 2.15, 20 °C, and 12 kV at normal polarity. The inter-assay CV was 7.6% for kynurenine and 6.8% for tryptophan.

#### 2.2.3. Homocysteine and Cysteine

Total plasma cysteine and homocysteine were measured using P/ACE Beckman capillary electrophoresis with laser-induced detection (Beckman Instruments, CA, USA), as described by Zinellu et al. [27]. Briefly, 100 µL of plasma sample was mixed with 10 µL of Tris(2-carboxyethyl)phosphine (200 mmol/L in water), vortexed for 30 s and then incubated at room temperature for 15 min. At the end of incubation, 100 µL of trichloroacetic acid (10%) was added, vortexed for 10s and then centrifuged at 3000× *g* for 10 min. Then, 100 µL of supernatant was mixed with 100 µL of 300 mmol/L Na_3_PO_4_ at pH 12.5 and 25 µL of 5-iodoacetamidofluorescein (4.1 mol/L). After incubation at room temperature for 10 min, samples were diluted 100-fold and injected into the capillary electrophoresis system. Analysis was performed in an uncoated fused-silica capillary was 75 µm ID and 57 cm length (50 cm to the detection window), applying 14 nL of sample and using 5 mmol/L sodium phosphate/ 4 mmol/L boric acid as electrolyte solution with 75 mmol/L N-methyl-D-glucamine at pH 11. The separating conditions (28 kV at normal polarity) were reached in 30 s and held at a constant voltage for 5 min. The separations were carried out at 40 °C. The inter-assay CV was 4.6% for homocysteine and 4.5% for cysteine.

#### 2.2.4. Taurine

Taurine was assessed by capillary electrophoresis with laser-induced detection as previously reported [28]. A volume of 50 µL of plasma was mixed with 50 µL internal standard homocysteic acid (200 µmol/L) and 100 µL of trichloroacetic acid (10%) was then added to precipitate the proteins. After centrifugation at 3000× *g* for 5 min, 10 µL of clear supernatant was mixed with 90 µL of 100 mmol/L Na_2_HPO_4_ of pH 9.5 and 11 µL of 15 mmol/L fluorescein isothiocyanate. After 20 min incubation time at 100 °C, the samples were diluted 100-fold and injected in capillary electrophoresis. A P/ACE 5510 capillary electrophoresis system equipped with a laser-induced fluorescence (LIF) detector (Beckman, CA, USA) was used. Analysis was performed in an uncoated fused silica capillary, 75 µm ID and 47 cm length (40 cm to the detection window), injecting 18 nL of sample. Separation was carried out in a 20 mmol/L tribasic sodium phosphate buffer, pH 11.8, 23 °C at normal polarity 22 kV. The inter-assay CV for taurine was 6.5%.

#### 2.2.5. Blood GSH

Blood GSH was measured by capillary electrophoresis with laser-induced detection as previously reported [29]. A total of 200 µL of blood were treated by adding 600 µL of cold water and keeping the samples at 4 °C for 15 min, then 200 µL of lysed samples were deproteinized by adding 200 µL acetonitrile and centrifuged at 2000× *g* for 5 min. The derivatization reaction was accomplished by mixing 100 µL supernatant with 100 µL 60 mmol/L sodium phosphate buffer, pH 12.5, and 25 µL of 4.1 mmol/L 5-IAF. After vortex-mixing, samples were incubated for 15 min at room temperature. Derivatized samples were diluted 100-fold in water and analyzed by capillary electrophoresis. A P/ACE 5510 capillary electrophoresis system equipped with laser-induced fluorescence was used (Beckman Instruments, CA, USA). The dimensions of the uncoated fused-silica capillary were 75 µm ID and 57 cm length (50 cm to the detection window). Analysis was performed applying 14 nL sample and using 5 mmol/L sodium phosphate, 4 mmol/L boric acid as electrolyte solution with 75 mmol/L N-methyl-D-glucamine. The separating conditions (30 kV, at normal polarity and 45 °C) were reached within 20 s and held at a constant voltage for 5 min. The inter-assay CV for blood GSH was <8%.

#### 2.2.6. TBARS

TBARS were determined according to the method described by Esterbauer and Cheeseman [30]. The assay measures MDA and other aldehydes produced by lipid peroxidation triggered by hydroxyl free radicals. Plasma was mixed with 10% trichloroacetic acid and 0.67% thiobarbituric acid and heated at 95 °C in a thermoblock heater for 25 min. A calibration curve was made using standard MDA and TBARS were determined by measuring the absorbance at 535 nm. The inter-assay CV for TBARS was <7%.

#### 2.2.7. PSH

Plasma protein -SH was spectrophotometrically assessed by using 5,5′-dithiobis-2-nitrobenzoic acid (DTNB) as titrating agent and by measuring the absorbance of conjugate at 405 nm [31]. Concentration in samples was determined from a GSH standard curve. Proteins –SH concentrations were normalized vs. protein plasma quantity measured by Lowry’s method. The inter-assay CV for TBARS was 8%.

#### 2.2.8. Nuclear Factor Erythroid 2-Related Factor 2 (Nrf2)

Plasma concentration of Nrf2 was assessed by commercial Elisa Kit (Antibodies-online.com, ABIN818803). The inter-assay CV for Nrf2 was <12%.

### 2.3. Statistical Analysis

The results are expressed as mean ± standard deviation (mean ± SD) or median and interquartile range (IQR). The distribution of variables was evaluated by means of Kolmogorov–Smirnov test for normal distribution. The statistical comparisons between groups were assessed by means of unpaired or paired Student’s t-test and Mann–Whitney test or Wilkoxon test, as appropriate. Correlations between variables were estimated using Pearson’s or Spearman’s correlation, as appropriate. Statistical analyses were performed using MedCalc for Windows, version 15.4 64 bit (MedCalc Software, Ostend, Belgium).

## 3. Results

Baseline demographic, clinical and biochemical characteristics of IPF patients and healthy controls are described in Table 1. As expected, both FVC (L) and FVC (%) were significantly lower in IPF patients than in controls (2.23 ± 0.66L vs 2.99 ± 0.84L, *p* = 0.005; 77.9 ± 20.1% vs 93.5 ± 12.9%, *p* = 0.009, respectively). There were no significant between-group differences in FEV_1_ (%) and BMI values. IPF patients had significantly lower concentrations of reduced blood GSH (457 ± 73 µmol/L vs 880 ± 212 µmol/L, *p* < 0.001) and plasma PSH (4.24 ± 0.95 µmol/g prot vs 5.28 ± 1.35 µmol/g prot, *p* = 0.012). By contrast, there were no significant between-group differences in Nrf2, Hcy, Cys, Tau, TBARS, Arginine, ADMA, ADMA/Arg ratio, Kyn, Trp, and Kyn/Trp ratio. Univariate regression analysis in IPF patients and healthy subjects as a combined group showed significant and positive relationships between FEV_1_ (%) and GSH (*r* = 0.39, *p* = 0.021), FVC (L) and GSH (*r* = 0.54, *p* = 0.0007), and FVC (L) and PSH (*r* = 0.42, *p* = 0.011).

Table 2 reports the clinical and biochemical characteristics of all IPF patients at baseline and after 24-week treatment with pirfenidone or nintedanib. Treatment was associated with a significant decrease in FVC (%) (77.9 ± 20.1% baseline vs. 74.4 ± 19.9% post-treatment) whilst no significant changes were observed in FEV_1_ (%), FVC (L), or DLCO (%). Furthermore, there was a significant treatment-induced increase in the concentrations of Tau (27.2 ± 7.2 μmol/L vs 33.5 ± 14.0 μmol/L, *p* = 0.037), PSH (4.24 ± 0.95 µmol/g prot vs 4.81 ± 1.32 µmol/g prot, *p* = 0.042), GSH (457 ± 73 μmol/L vs 619 ± 222 μmol/L, *p* = 0.006) and a reduction in Kyn (1.38 ± 0.49 μmol/L vs. 1.17 ± 0.50 μmol/L, *p* = 0.047). A non-significant trend toward a decrease of TBARS concentrations was observed (4.01 ± 1.71 μmol/L vs 3.32 ± 1.17 μmol/L, *p* = 0.08) while the remaining biomarkers were substantially unchanged.

Pirfenidone treatment (Table 3) did not significantly change FVC (%), FVC (L), DLCO (%) and FEV_1_ (L). However, it significantly reduced the Kyn/Trp ratio (0.030 ± 0.011 vs 0.025 ± 0.010, *p* = 0.048). Non-significant trends were also observed toward an increase in PSH (+9%), GSH (+23%), and Tau (+23%), and a decrease in TBARS (−17%) (Figure 1).

Similarly, nintedanib treatment (Table 4) did not significantly affect lung functional parameters. However, it significantly increased blood GSH (486 ± 70 μmol/L vs 723 ± 194 μmol/L, *p* = 0.006) and reduced ADMA (0.501 ± 0.094 μmol/L vs 0.468 ± 0.071 μmol/L, *p* = 0.024) concentrations. Non-significant trends toward an increase in PSH (+19%) and Tau (+23%) and a decrease in Kyn/Trp ratio (−21%) and TBARS (−18%) were also observed.

## 4. Discussion

For the present study we a priori selected the following markers, on the basis of experimental evidence in lung diseases, as biomarkers of oxidative stress: TBARS, GSH, PSH, Nrf2, Hcy, Cys, and ADMA; or inflammation: kynurenine (Kyn), tryptophan (Trp) and Kyn/Trp ratio.

Nrf2 is a transcription factor that controls the expression of a group of antioxidant and detoxification enzymes and pulmonary proteins responsible for cellular defence and survival against OS. During oxidative stress Nrf2 translocates from the cytosol into the nucleus where, after binding to antioxidant response elements, it induces the transcription of antioxidant genes [32]. Reduction of Nrf2 signaling is associated with an increased susceptibility to insults deriving from OS in humans and animals [33]. Thiobarbituric acid-reactive substances originate from oxidative alteration of polyunsaturated fatty acids. Decomposition of lipids results in the formation of aldehydes, principally malondialdehyde, which can be colorimetrically quantified by thiobarbituric acid reaction. TBARS detection is a widely used method for the screening and monitoring of lipid peroxidation [34]. GSH is a tripeptide thiol concentrated principally into the cells that reduces ROS such as superoxide or the hydroxyl radical. During oxidative reactions, a decrease in reduced GSH concentrations occurs due to its conversion to glutathione disulphide (GSSG). Reduced glutathione is then restored by specific glutathione reductases that have high affinity for GSSG and, particularly, NADPH. An excess in OS may alter the normal balance between GSH and GSSG leading to the overproduction of oxidised form and GSH decreases [35]. Taurine is a small sulphur-containing amino acid found in high amounts, millimolar concentrations, in mammal tissues and in micromolar concentrations in body fluids. It plays different physiological roles in several tissues as a neurotransmitter, osmolyte and antioxidant. Although is well known that taurine is unable to directly scavenge free radicals, several lines of evidence suggest that it is an effective inhibitor of ROS generation. In particular, it prevents the leakage of the reactive compounds formed in the mitochondrial environment by stabilizing electron chain transport, thus acting as an antioxidant [36,37]. PSH refers to total protein -SH groups. Given the elevated concentration of albumin, the free cysteinyl thiol of this protein is the most abundant in human plasma. The -SH group of albumin Cys^34^ accounts for about 80% of reduced thiols in plasma and it is an important scavenger of reactive oxygen and nitrogen species, acting as a key redox buffer in blood. During oxidative stress, -SH groups become oxidized to disulphide or sulfenic acid, leading to -SH decrease. Homocysteine is a non-proteinogenic amino acid linked with several major disease states. Hcy can cyclize to give homocysteine thiolactone, a five-membered heterocycle. Subsequent cleavage of this molecule during protein N-homocysteinylation reaction may generate reactive species [38]. In addition, elevated Hcy concentrations inhibit mitochondrial respiration resulting in ROS accumulation [39]. Albeit less reactive than homocysteine, cysteine exhibits some of the chemical properties of Hcy, due to the presence of the sulfhydryl group in the molecule. Cysteine demonstrates cytotoxicity in vitro and shows auto-oxidation properties in the presence of metal ions, resulting in the generation of free radicals and hydrogen peroxide. The methylated arginine ADMA is an endogenous inhibitor of nitric oxide synthase synthetized by protein arginine N-methyltransferases (PRMTs) through the addition of two methyl groups to the terminal nitrogen atom of arginine contained in proteins. Several studies have shown that RNA or protein expression of PRMT1 is increased, and DDAH activity is decreased, under OS stimuli [40]. The kynurenine pathway is the major route of degradation of the essential amino acid tryptophan. The plasma kynurenine to tryptophan ratio is frequently used to express the activity of the extrahepatic Trp-degrading enzyme indoleamine 2,3-dioxygenase (IDO). Degradation of Try by extrahepatic IDO, although not significant in physiological conditions, markedly rises during immune activation, infections or OS. Recently, Kyn and the Kyn/Trp ratio have been proposed as sensitive biomarkers of systemic inflammation in COPD patients [41].

OS, defined as an imbalance between oxidant production and antioxidant defence in favour of oxidants, leading to cellular and tissue injury, plays a significant role in IPF [11,12]. The lungs are continuously exposed to exogenous oxidants (air pollutants, cigarette smoke, etc.) that further increase endogenous oxidants production through activation of inflammatory cells to generate free radicals. In addition, different pathways can generate ROS in human lungs, including mitochondrial electron transport chain, nicotinamide adenine dinucleotide phosphate oxidases, xanthine oxidase, myeloperoxidase and eosinophil peroxidase [42,43]. Moreover, superoxide may react with NO to form reactive nitrogen species (RNS), such as peroxynitrite. NO is widely produced by the inducible form of nitric oxide synthase (iNOS, NOS2) in the lung, particularly during inflammation. Human lung cells, besides, largely express also the constitutive isoforms of NOS, that contribute to NO production for the maintenance of tissue integrity. Therefore, an appropriate balance between intracellular and extracellular oxidants and antioxidants is required for the maintenance of pulmonary homeostasis. Lung protection against oxidants is guaranteed by protective antioxidant proteins and molecules such as low-molecular-weight antioxidants (glutathione, vitamins, uric acid), mucins, metal-binding proteins (transferrin, lactoferrin, metallothionein), enzymes able to reduce H_2_O_2_ (some glutathione-associated enzymes and catalase), intracellular and extracellular superoxide dismutases, enzymes involved in detoxification processes (glutathione-S-transferases), and other redox regulatory thiol proteins (thioredoxin-peroxiredoxin system and glutaredoxins) [12]. In line with the available evidence on the role of OS, we found decreased concentrations of thiol groups in IPF patients, resulting from a reduction in both GSH concentrations in red blood cells (−48%) and PSH plasma concentrations (−20%). These data agree with a previous report by Veith et al. which showed significant reductions in blood GSH concentrations (50%) and total plasma antioxidant capacity in IPF patients when compared to controls [44]. Similarly, Beeh et al. [45] reported that the total GSH concentrations in IPF patients were about four-fold lower than those in controls. Furthermore, there was an inverse association between GSH sputum concentrations and disease severity, and a positive correlation between GSH and vital capacity. These data agree with our finding of a positive correlation between GSH and the lung function parameters, FVC (%) and FEV1 (%). A halving of GSH concentrations in epithelial lining fluid (ELF) has also been described by Meyer et al. [46]. In the same matrix, Cantin et al. showed that the concentration of total GSH in IPF patients was less than 75% of that measured in healthy controls [47]. Interestingly, the same authors reported that, despite the differences in the total content of GSH, the redox status [GSH]/[GSH + GSSG] in patients with IPF was similar to that observed in normal subjects. Assuming that ELF GSH is derived from epithelial cells, the authors hypothesized that the chronic pro-oxidant burden in IPF impairs their ability to synthesize, store and secrete GSH. This hypothesis is supported by studies reporting that oxidative stress may reduce GSH concentrations in neutrophils and lung tissue and affect GSH transport in epithelial cells [48,49].

Contrary to the results of a study by Rahaman et al. [50], that found increased plasma TBARS concentrations in IPF patients compared to healthy subjects, we found similar concentrations of lipid peroxidation products in IPF patients and controls. This discrepancy may be related, at least in part, to the different disease severity of the IPF patients studied. In particular, our patients generally exhibited mild disease (mainly in stage I and, to a lesser extent, in stage II) and it is possible that, in this group, the oxidative burden is still efficiently controlled by antioxidants action. In this context, Bartoli et al. also did not observe significant differences in TBARS concentrations in exhaled breath condensate between IPF patients and healthy subjects [51].

Few reports have described the measurement of Nrf2 in an extracellular fluid. However, the feasibility of a plasma assay of Nrf2 has been demonstrated by Ban WH et al. [52], who reported that plasma Nrf2 concentrations gradually increased with disease severity and the extent of systemic inflammation in patients with COPD, and by Sireesh D. et al. [53], who found that plasma Nrf2 concentrations were lower in patients with type 2 diabetes when compared to control subjects.

We did not find significant differences in Nrf2 plasma concentrations between IPF patients and controls. This agrees with previous observations by Mazur et al. who failed to detect significant differences in Nrf2 concentrations in lung tissue between seven IPF patients and seven healthy subjects [54]. Interestingly, the same authors found, by immunohistochemistry analysis of IPF patient tissues, that Nrf2 significantly increased in the hyperplastic alveolar epithelium compared to the normal alveolar epithelium, and that Nrf2 was markedly expressed in the nuclear compartment of the hyperplastic cells. It has been suggested that this Nrf2 upregulation might represent a failed attempt to restore the pulmonary redox status, thus leaving IPF patients exposed to elevated ROS concentrations [55].

We assessed, for the first time, homocysteine, taurine, ADMA, ADMA/arginine ratio, kynurenine and kynurenine/tryptophan ratio in IPF patients however there were no significant differences in the plasma concentrations of these analytes between IPF patients and controls. While the relative mild disease severity in our patients combined to the relatively small sample size might at least partly explain the lack of significant differences observed, the non-significant (*p* = 0.06) trend toward an increase of kynurenine concentrations in IPF patients suggests the presence of a pro-inflammation status in this group.

In recent years, two anti-fibrotic drugs, nintedanib and pirfenidone, have been approved for the treatment of IPF. Although their mechanisms of action are not fully characterized, pirfenidone is thought to inhibit TGF-β-mediating signaling pathways, thus decreasing fibroblast proliferation and collagen synthesis [16]. In vitro studies have shown that nintedanib inhibits fundamental processes involved in fibrosis, particularly the recruitment, proliferation and differentiation of fibroblasts and fibrocytes and extracellular matrix deposition, through the inhibition of signaling pathways mediated by tyrosine kinases [18]. A retardation in lung function decline and a decreased risk of acute respiratory deteriorations has been described with both drugs. Although clinical trials have failed to demonstrate a significant improvement in IPF symptoms, antifibrotic therapies improve life expectancy [56]. In addition, both pirfenidone and nintedanib have been shown to exert significant anti-inflammatory and antioxidant effects in vitro however human data are lacking [16,17,19,20]. In our study, 24-week treatment with pirfenidone or nintedanib, when collectively analysed in the whole IPF cohort, significantly increased plasma taurine, GSH and PSH concentrations and reduced kynurenine concentrations. A non-significant (*p* = 0.08) trend toward TBARS reduction was also observed. When analysing specific anti-fibrotic agents, pirfenidone treatment significantly reduced the Kyn/Trp ratio. Furthermore, a non-significant trend toward an increase in GSH and PSH, and a decrease in TBARS, was also observed. These data are in agreement with previous in vitro studies that showed that pirfenidone can downregulate proinflammatory cytokines, such as TNF-α, IFN-γ, and interleukin (IL)-6, and improve survival in a murine model of lipopolysaccharide-induced acute lung injury. In vitro studies in human peripheral blood lymphocytes and mice studies demonstrated the ability of pirfenidone to exert similar anti-inflammatory effects by inhibiting staphylococcal enterotoxin B-induced proliferation and the synthesis of TNF-α, IFN-γ, IL-1β, and IL-6 [55]. With regard to antioxidant properties, pirfenidone has been reported to reduce ROS production [17], act as a scavenger of free hydroxyl and superoxide anion, and block NADPH-dependent lipid peroxidation in a concentration-dependent manner [16]. In nintedanib treated patients we found a significant reduction in ADMA concentrations, an increase in blood GSH concentrations and a non-significant trend toward an increase in PSH concentrations. These results are also in agreement with previous reports that showed that nintedanib significantly reduces ROS production and transglutaminase-2 expression, both in vivo and in vitro [19]. A borderline significant increase in taurine concentrations was also observed with individual anti-fibrotic agents. This increase was significant when the two treatment groups were considered together, suggesting that taurine might also serve as a marker of antioxidant effects during anti-fibrotic treatment in IPF. It has been reported that the antioxidant activity of taurine is linked to improved mitochondrial function, which reduces mitochondrial superoxide generation. In particular, taurine is a key regulator of mitochondrial protein synthesis, improving electron transport chain efficiency and limiting superoxide generation thus protecting the mitochondria [36]. There is also experimental evidence that taurine treatment significantly suppresses the bleomicin-induced increase in malondialdehyde concentration, superoxide dismutase activity and collagen accumulation in lung hamster, completely or partially ameliorating chemically-induced pulmonary fibrosis [56]. In addition, both nintedanib and, to a lesser extent, pirfenidone increase blood GSH concentrations in IPF. This is likely the result of enhanced synthesis and storing of GSH which, as reported by Cantin et al. [57], are both impaired in IPF due to chronic oxidative burden. The treatment-induced increase in GSH concentrations might be clinically relevant also in consideration of the described association between GSH concentrations and lung functional parameters and the fact that GSH has been proposed to play a critical role in the anti-fibrotic response [58].

## 5. Conclusions

In conclusion, despite its preliminary nature and the relatively small sample size, this study has reported for the first time the effects of pirfenidone and nintedanib on a comprehensive panel of markers of inflammation and OS in IPF patients. All reported data, including non-significant differences, may assist with the adequate design (i.e., sample size and/or biomarker selection) of larger interventional trials to confirm these findings and to establish whether such effects are associated with clinical improvement in this group.

## Figures and Tables

**Figure 1 antioxidants-09-01064-f001:**
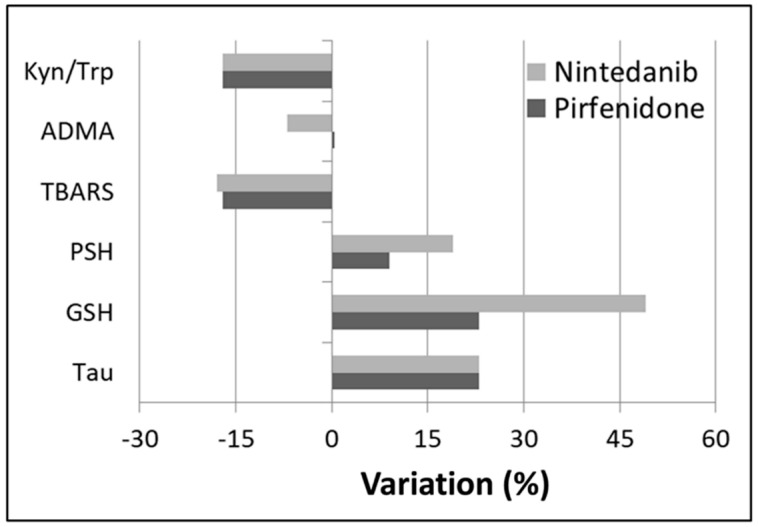
Most relevant percentage variation of biochemical parameters after 24 weeks of drug administration.

**Table 1 antioxidants-09-01064-t001:** Demographic, clinical and biochemical characteristics of healthy subjects and IPF patients.

	Controls (*n* = 18)	IPF (*n* = 18)	*p* Value
Age (years)	67.7 ± 6.7	66.7 ± 10.3	0.75
Gender (F/M)	8/10	8/10	1
BMI (kg/m^2^)	26.7 ± 5.0	28.9 ± 4.7	0.19
Smokers, yes/no	14/4	14/4	1
FVC (L)	2.99 ± 0.84	2.23 ± 0.66	**0.005**
FVC (% predicted)	93.5 ± 12.9	77.9 ± 20.1	**0.009**
FEV1(% predicted)	94.4 ± 17.2	87.2 ± 23.2	0.29
DLCO (% predicted)	--	52.1 ± 16.8	--
GAP index, (1/2/3/4/5/6/7 n)	--	(2/3/7/3/3/0/0)	--
Disease stage, (I/II/III n)	--	(11/7/0)	--
Hcy, (μmol/L)	15.4 ± 5.7	17.8 ± 5.6	0.21
Cys, (μmol/L)	340 ± 59	353 ± 54	0.50
Tau, (μmol/L)	28.0 ± 7.8	27.2 ± 7.2	0.75
GSH, (μmol/L)	880 ± 212	457 ± 73	**<0.001**
PSH, (µmol/g prot)	5.28 ± 1.35	4.24 ± 0.95	**0.012**
TBARS, (μmol/L)	3.92 ± 0.79	4.01 ± 1.1.71	0.85
Nrf2, (pg/mL)	80 (50-107)	100 (61-172)	0.12
Arginine, (μmol/L)	52.0 ± 11.6	50.8 ± 10.6	0.74
ADMA, (μmol/L)	0.433 ± 0.075	0.464 ± 0.091	0.27
ADMA/Arg ratio	0.0086 ± 0.0020	0.0095 ± 0.0029	0.29
Kynurenine, (μmol/L)	1.06 ± 0.50	1.38 ± 0.49	0.06
Tryptophan, (μmol/L)	42.3 ± 17.7	44.4 ± 15.5	0.70
Kyn/Trp ratio	0.027 ± 0.013	0.032 ± 0.012	0.18

Data are presented as mean ± standard deviation or median (IQR); FEV1: Forced Expiratory Volume in the 1st second; FVC: Forced Vital Capacity; DLCO, Diffusion *Lung* Carbon Oxide. Bold values indicate statistical significance at the *p* < 0.05 level.

**Table 2 antioxidants-09-01064-t002:** Clinical and laboratory characteristics of IPF patients pre- and post- treatment.

	Pre-Treatment (*n* = 18)	Post-Treatment (*n* = 18)	*p* Value
FVC (L)	2.23 ± 0.66	2.14 ± 0.67	0.07
FVC (% predicted)	77.9 ± 20.1	74.4 ± 19.9	**0.04**
FEV1(% predicted)	87.2 ± 23.2	84.8 ± 23.8	0.15
DLCO (% predicted)	52.1 ± 16.8	47.3 ± 22.6	0.08
Hcy, (μmol/L)	17.8 ± 5.6	18.4 ± 6.0	0.72
Cys, (μmol/L)	353 ± 54	346 ± 47	0.53
Tau, (μmol/L)	27.2 ± 7.2	33.5 ± 14.0	**0.037**
GSH, (μmol/L)	457 ± 73	619 ± 222	**0.006**
PSH, (µmol/g prot)	4.24 ± 0.95	4.81 ± 1.32	**0.042**
TBARS, (μmol/L)	4.01 ± 1.71	3.32 ± 1.17	0.08
Erf2, (pg/mL)	100 (61-172)	100 (95-129)	0.77
Arginine, (μmol/L)	50.8 ± 10.6	50.8 ± 12.6	0.99
ADMA, (μmol/L)	0.464 ± 0.091	0.448 ± 0.072	0.13
ADMA/Arg ratio	0.0095 ± 0.0029	0.0094 ± 0.0032	0.75
Kynurenine, (μmol/L)	1.38 ± 0.49	1.17 ± 0.50	**0.047**
Tryptophan, (μmol/L)	44.4 ± 15.5	45.4 ± 8.9	0.79
Kyn/Trp ratio	0.032 ± 0.012	0.027 ± 0.014	0.11

Data are presented as mean ± standard deviation or median (IQR); FEV1: Forced Expiratory Volume in the 1st second; FVC: Forced Vital Capacity; DLCO, Diffusion *Lung* Carbon Oxide. Bold values indicate statistical significance at the *p* < 0.05 level.

**Table 3 antioxidants-09-01064-t003:** Clinical and biochemical characteristics of IPF patients at baseline and after 24 weeks of pirfenidone treatment.

	Pre-Treatment (*n* = 10)	Post-Treatment (*n* = 10)	*p* Value
FVC (L)	2.46 ± 0.73	2.37 ± 0.75	0.17
FVC (% predicted)	80.7 ± 21.1	77.1 ± 22.6	0.18
FEV1(% predicted)	89.9 ± 25.3	89.1 ± 26.7	0.71
DLCO (% predicted)	54.7 ± 19.4	50.9 ± 24.4	0.26
Hcy, (μmol/L)	19.3 ± 6.2	18.2 ± 5.4	0.40
Cys, (μmol/L)	350 ± 63	330 ± 47	0.13
Tau, (μmol/L)	28.6 ± 8.0	35.1 ± 17.0	0.16
GSH, (μmol/L)	435 ± 70	535 ± 215	0.21
PSH, (µmol/g prot)	4.36 ± 1.10	4.75 ± 1.31	0.30
TBARS, (μmol/L)	3.94 ± 1.82	3.28 ± 0.93	0.17
Nrf2, (pg/mL)	115 (47-211)	98 (95-165)	0.97
Arginine, (μmol/L)	51.8 ± 10.8	53.1 ± 15.0	0.82
ADMA, (μmol/L)	0.434 ± 0.080	0.432 ± 0.072	0.90
ADMA/Arg ratio	0.0087 ± 0.0024	0.0089 ± 0.0037	0.79
Kynurenine, (μmol/L)	1.33 ± 0.53	1.14 ± 0.35	0.14
Tryptophan, (μmol/L)	45.1 ± 19.7	47.7 ± 10.0	0.63
Kyn/Trp ratio	0.030 ± 0.011	0.025 ± 0.010	**0.048**

Data are presented as mean ± standard deviation or median (IQR); FEV1: Forced Expiratory Volume in the 1st second; FVC: Forced Vital Capacity; DLCO, Diffusion *Lung* Carbon Oxide. Bold values indicate statistical significance at the *p* < 0.05 level.

**Table 4 antioxidants-09-01064-t004:** Clinical and laboratory characteristics of IPF patients at baseline and after 24 weeks of Nintedanib treatment.

	Pre-Treatment (*n* = 8)	Post-Treatment (*n* = 8)	*p* Value
FVC (L)	1.95 ± 0.47	1.85 ± 0.31	0.29
FVC (% predicted)	74.4 ± 19.6	71.0 ± 16.6	0.14
FEV1(% predicted)	83.7 ± 21.4	79.4 ± 20.0	0.10
DLCO (% predicted)	48.8 ± 13.3	42.8 ± 20.6	0.23
Hcy, (μmol/L)	15.9 ± 4.3	18.7 ± 7.1	0.42
Cys, (μmol/L)	358 ± 45	366 ± 41	0.72
Tau, (μmol/L)	25.5 ± 6.3	31.4 ± 9.9	0.14
GSH, (μmol/L)	486 ± 70	723 ± 194	**0.006**
PSH, (µmol/g prot)	4.10 ± 0.77	4.89 ± 1.43	0.08
TBARS, (μmol/L)	4.10 ± 1.68	3.37 ± 1.50	0.31
Nrf2, (pg/mL)	84 (67-152)	105 (81-129)	0.64
Arginine, (μmol/L)	49.4 ± 10.7	47.9 ± 8.8	0.44
ADMA, (μmol/L)	0.501 ± 0.094	0.468 ± 0.071	**0.024**
ADMA/Arg ratio	0.0106 ± 0.0032	0.0100 ± 0.0026	0.30
Kynurenine, (μmol/L)	1.46 ± 0.48	1.21 ± 0.67	0.21
Tryptophan, (μmol/L)	43.4 ± 9.1	42.4 ± 6.8	0.84
Kyn/Trp ratio	0.035 ± 0.013	0.029 ± 0.017	0.43

Data are presented as mean ± standard deviation or median (IQR); FEV1: Forced Expiratory Volume in the 1st second; FVC: Forced Vital Capacity; DLCO, Diffusion *Lung* Carbon Oxide. Bold values indicate statistical significance at the *p* < 0.05 level.

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
