# Peer review of "Effects of Pirfenidone and Nintedanib on Markers of Systemic Oxidative Stress and Inflammation in Patients with Idiopathic Pulmonary Fibrosis: A Preliminary Report"

_antioxidants, 2020, doi:10.3390/antiox9111064_

Round 1
Reviewer 1 Report
The study by Fois et al. aimed to evaluate the effects of two drugs (i.e. pirfenidone and nintedanib) for the treatment of idiopathic pulmonary fibrosis on systemic redox balance. To address this objective the authors measured a number of putative biomarkers of oxidative damage and antioxidant defense (besides some not conventional markers of inflammation) at baseline and after 24-week treatment in 18 patients affected by idiopathic pulmonary fibrosis (10 treated with 25 pirfenidone and 8 with nintedanib) and in healthy controls.
In my opinion the manuscript has some merits: it is well-written, clear and easy to follow also for readers that are not familiar with the topic. However, the work also suffers from severe methodological shortcomings that limits its overall scientific quality .
- The main issues of the study are due to the markers used for measuring the systemic redox homeostasis. None of the (several) molecules measured by the authors can be considered as reliable and analytically accurate markers for oxidative stress.
The choice of the authors can, at least in part, be justified by the absence of universally accepted “golden-standard” biomarker for the in vivo assessment of oxidative stress. However, according to some articles from major expert in redox biology field [1–6], there are very good alternatives such as F2-isoprostanes (as assessed by GS/MS), MDA (but only determined by HPLC plus florescence) and 8-oxo-2'-deoxyguanosine (8-oxo-dG) [2,4,7,8].
Regarding some markers considered in the study, there are some considerations to make (and always report in the manuscript as evident intrinsic methodological limitations): 1) TBARS is widely considered as a highly unspecific biomarker of lipid peroxidation (“TBA can react with several compounds, including sugars, amino acids, bilirubin, and albumin, producing interferences in the measurement”[9]) 2) Homocysteine is not a direct marker of oxidative damage and its concentration is influenced by a wider range of other endogen/exogen factors 3) GSH and thiols are not biomarkers of antioxidant power, especially when they are measured in plasma/serum, because of their great chemical instability. The ratio GSH/GSSG is a good way to measure the degree of oxidative stress in a cell, but not in the more complex and variegate environment of blood.
- The authors should not describe the biological significance and the function of the markers in the “Methods” section. This description can be made in the “Discussion” or, briefly, in the “Introduction”.
- The (very) small size of the sample limits the validity and reliability of the statistical analysis
- The role of NRF2 is crucial in the response to oxidative insult. However, there are just a few research papers dealing with the measurement of this transcription factors in an extracellular fluid. The authors should justify and support with published evidence the use of this molecule as peripheral biomarkers of antioxidant defensive system.
- Kalyanaraman, B.; Darley-Usmar, V.; Davies, K.J.A.; Dennery, P.A.; Forman, H.J.; Grisham, M.B.; Mann, G.E.; Moore, K.; Roberts, L.J.; Ischiropoulos, H. Measuring reactive oxygen and nitrogen species with fluorescent probes: Challenges and limitations. Free Radic. Biol. Med. 2012, 52, 1–6, doi:10.1016/j.freeradbiomed.2011.09.030.
- Kadiiska, M.B.; Gladen, B.C.; Baird, D.D.; Germolec, D.; Graham, L.B.; Parker, C.E.; Nyska, A.; Wachsman, J.T.; Ames, B.N.; Basu, S.; et al. Biomarkers of oxidative stress study II: are oxidation products of lipids, proteins, and DNA markers of CCl4 poisoning? Free Radic. Biol. Med. 2005, 38, 698–710, doi:10.1016/j.freeradbiomed.2004.09.017.
- Kadiiska, M.B.; Peddada, S.; Herbert, R.A.; Basu, S.; Hensley, K.; Jones, D.P.; Hatch, G.E.; Mason, R.P. Biomarkers of oxidative stress study VI. Endogenous plasma antioxidants fail as useful biomarkers of endotoxin-induced oxidative stress. Free Radic. Biol. Med. 2015, 81, 100–6, doi:10.1016/j.freeradbiomed.2015.01.006.
- Dalle-Donne, I.; Rossi, R.; Colombo, R.; Giustarini, D.; Milzani, A. Biomarkers of oxidative damage in human disease. Clin. Chem. 2006, 52, 601–623, doi:10.1373/clinchem.2005.061408.
- Forman, H.J.; Davies, K.J.A.; Ursini, F. How do nutritional antioxidants really work: nucleophilic tone and para-hormesis versus free radical scavenging in vivo. Free Radic. Biol. Med. 2014, 66, 24–35, doi:10.1016/j.freeradbiomed.2013.05.045.
- Jay, H.; Augusto, O.; Brigelius-flohe, R.; Dennery, P.A.; Kalyanaraman, B.; Ischiropoulos, H.; Mann, G.E.; Radi, R.; Roberts, L.J.; Vina, J.; et al. Even free radicals should follow some rules : A Guide to free radical research terminology and methodology. Free Radic. Biol. Med. 2015, 78, 233–235, doi:10.1016/j.freeradbiomed.2014.10.504.
- Klawitter, J.; Haschke, M.; Shokati, T.; Klawitter, J.; Christians, U. Quantification of 15-F2t-isoprostane in human plasma and urine: results from enzyme-linked immunoassay and liquid chromatography/tandem mass spectrometry cannot be compared. Rapid Commun. Mass Spectrom. 2011, 25, 463–468, doi:10.1002/rcm.4871.
- Milne, G.L.; Gao, B.; Terry, E.S.; Zackert, W.E.; Sanchez, S.C. Measurement of F2- isoprostanes and isofurans using gas chromatography-mass spectrometry. Free Radic. Biol. Med. 2013, 59, 36–44, doi:10.1016/j.freeradbiomed.2012.09.030.
- Marrocco, I.; Altieri, F.; Peluso, I. Measurement and Clinical Significance of Biomarkers of Oxidative Stress in Humans. Oxid. Med. Cell. Longev. 2017, 2017, 1–32, doi:10.1155/2017/6501046.
Author Response
The study by Fois et al. aimed to evaluate the effects of two drugs (i.e. pirfenidone and nintedanib) for the treatment of idiopathic pulmonary fibrosis on systemic redox balance. To address this objective the authors measured a number of putative biomarkers of oxidative damage and antioxidant defense (besides some not conventional markers of inflammation) at baseline and after 24-week treatment in 18 patients affected by idiopathic pulmonary fibrosis (10 treated with 25 pirfenidone and 8 with nintedanib) and in healthy controls.
In my opinion the manuscript has some merits: it is well-written, clear and easy to follow also for readers that are not familiar with the topic. However, the work also suffers from severe methodological shortcomings that limits its overall scientific quality.
- The main issues of the study are due to the markers used for measuring the systemic redox homeostasis. None of the (several) molecules measured by the authors can be considered as reliable and analytically accurate markers for oxidative stress. The choice of the authors can, at least in part, be justified by the absence of universally accepted “golden-standard” biomarker for the in vivo assessment of oxidative stress. However, according to some articles from major expert in redox biology field [1–6], there are very good alternatives such as F2-isoprostanes (as assessed by GS/MS), MDA (but only determined by HPLC plus florescence) and 8-oxo-2'-deoxyguanosine (8-oxo-dG) [2,4,7,8].Regarding some markers considered in the study, there are some considerations to make (and always report in the manuscript as evident intrinsic methodological limitations): 1) TBARS is widely considered as a highly unspecific biomarker of lipid peroxidation (“TBA can react with several compounds, including sugars, amino acids, bilirubin, and albumin, producing interferences in the measurement”[9]) 2) Homocysteine is not a direct marker of oxidative damage and its concentration is influenced by a wider range of other endogen/exogen factors 3) GSH and thiols are not biomarkers of antioxidant power, especially when they are measured in plasma/serum, because of their great chemical instability. The ratio GSH/GSSG is a good way to measure the degree of oxidative stress in a cell, but not in the more complex and variegate environment of blood.
We agree with the Reviewer that more reliable and analytically accurate biomarkers could be used to evaluate OS. However this primarily depends on access to highly complex instruments such as GS/MS or HPLC-Flu, which we do not have. That said, as also highlighted by the Reviewer, there is no universally accepted “golden-standard” biomarker for the in vivo assessment of OS. We acknowledge the limitations of the TBARS assay, widely discussed in our previous reports [1-4]. However, we have also demonstrated the utility of this assay in evaluating OS in several diseases [5-12].
We also agree that homocysteine is not a direct marker of oxidative damage. In fact, we state “…Hcy can cyclize to give homocysteine thiolactone, a five-membered heterocycle. Subsequent cleavage of this molecule during protein N-homocysteinylation reaction may generate reactive species [13]. In addition, elevated Hcy concentrations inhibit mitochondrial respiration resulting in ROS accumulation [14]” (page 9, lines 270-273 ). Thus, Hcy plasma concentrations may be associated with OS [15-17].
Although we agree in principle with the comment on GSH, its measurement in blood has often been reported in the literature as an approach to evaluate OS. For example, 22 studies investigating GSH as an OS biomarker have been published in relation to a single condition such as chronic obstructive pulmonary disease [18-39].
- The authors should not describe the biological significance and the function of the markers in the “Methods” section. This description can be made in the “Discussion” or, briefly, in the “Introduction”.
We thank the Reviewer for the suggestion. This section has been moved to the discussion.
- The (very) small size of the sample limits the validity and reliability of the statistical analysis
We agree with the Reviewer. This is reflected in the title, “preliminary report”, and study limitations (page 11, lines 397-399).
- The role of NRF2 is crucial in the response to oxidative insult. However, there are just a few research papers dealing with the measurement of this transcription factors in an extracellular fluid. The authors should justify and support with published evidence the use of this molecule as peripheral biomarkers of antioxidant defensive system.
We thank the Reviewer for the suggestion. Accordingly, several sentences and references have been added in the discussion to provide additional evidence regarding the use of this molecule as a peripheral biomarker of the antioxidant defensive system (page 10 lines 330-334).
- Pinna A, Boscia F, Paliogiannis P, Carru C, Zinellu A. Malondialdehyde levels in patients with age-related macular degeneration: A Systematic Review and Meta-analysis. Retina. 2020 Feb;40(2):195-203.
- Zinellu A, Paliogiannis P, Usai MF, Carru C, Mangoni AA. Effect of statin treatment on circulating malondialdehyde concentrations: a systematic review and meta-analysis. Ther Adv Chronic Dis. 2019 Jul 18;10:2040622319862714.
- Paliogiannis P, Fois AG, Sotgia S, Mangoni AA, Zinellu E, Pirina P, Carru C, Zinellu A. Circulating malondialdehyde concentrations in patients with stable chronic obstructive pulmonary disease: a systematic review and meta-analysis. Biomark Med. 2018 Jul;12(7):771-781.
- Paliogiannis P, Fois AG, Collu C, Bandinu A, Zinellu E, Carru C, Pirina P, Mangoni AA, Zinellu A. Oxidative stress-linked biomarkers in idiopathic pulmonary fibrosis: a systematic review and meta-analysis. Biomark Med. 2018 Oct;12(10):1175-1184.
- Bassu S, Zinellu A, Sotgia S, Mangoni AA, Floris A, Farina G, Passiu G, Carru C, Erre GL. Oxidative Stress Biomarkers and Peripheral Endothelial Dysfunction in Rheumatoid Arthritis: A Monocentric Cross-Sectional Case-Control Study. 2020 Aug 25;25(17):E3855.
- Gambino CM, Accardi G, Aiello A, Caruso C, Carru C, Gioia BG, Guggino G, Rizzo S, Zinellu A, Ciaccio M, Candore G. Uncoupling protein 2 as genetic risk factor for Systemic Lupus Erythematosus: association with malondialdehyde levels and intima media thickness. Minerva Cardioangiol. 2020 Jun 1. doi: 10.23736/S0026-4725.20.05225-1.
- Carru C, Da Boit M, Paliogiannis P, Zinellu A, Sotgia S, Sibson R, Meakin JR, Aspden RM, Mangoni AA, Gray SR. Markers of oxidative stress, skeletal muscle mass and function, and their responses to resistance exercise training in older adults. Exp Gerontol. 2018 Mar;103:101-106. doi: 10.1016/j.exger.2017.12.024.
- Zinellu A, Sotgia S, Sotgiu E, Assaretti S, Baralla A, Mangoni AA, Satta AE, Carru C. Cholesterol lowering treatment restores blood global DNA methylation in chronic kidney disease (CKD) patients. Nutr Metab Cardiovasc Dis. 2017 Sep;27(9):822-829.
- Zinellu A, Sotgia S, Mangoni AA, Sotgiu E, Ena S, Satta AE, Carru C. Effect of cholesterol lowering treatment on plasma markers of endothelial dysfunction in chronic kidney disease. J Pharm Biomed Anal. 2016 Sep 10;129:383-388.
- Zinellu A, Sotgia S, Mangoni AA, Sanna M, Satta AE, Carru C. Impact of cholesterol lowering treatment on plasma kynurenine and tryptophan concentrations in chronic kidney disease: relationship with oxidative stress improvement. Nutr Metab Cardiovasc Dis. 2015 Feb;25(2):153-9.
- Campesi I, Carru C, Zinellu A, Occhioni S, Sanna M, Palermo M, Tonolo G, Mercuro G, Franconi F. Regular cigarette smoking influences the transsulfuration pathway, endothelial function, and inflammation biomarkers in a sex-gender specific manner in healthy young humans. Am J Transl Res. 2013 Aug 15;5(5):497-509.
- Zinellu A, Sotgia S, Loriga G, Deiana L, Satta AE, Carru C. Oxidative stress improvement is associated with increased levels of taurine in CKD patients undergoing lipid-lowering therapy. Amino Acids. 2012 Oct;43(4):1499-507.
- Sibrian-Vazquez, M.; Escobedo, J.O.; Lim, S.; Samoei, G.K.; Strongin, R.M. Homocystamides promote free-radical and oxidative damage to proteins. Proc. Natl. Acad. Sci. USA 2010, 107, 551-554.
- Folbergrová. J.; Jesina, P.; Drahota, Z.; Lisý, V.; Haugvicová, R.; Vojtísková, A.; HoustÄ•k, J. Mitochondrial complex I inhibition in cerebral cortex of immature rats following homocysteic acid-induced seizures. Exp. Neurol. 2007, 204, 597-609.
- Zhang X, Huang Z, Xie Z, Chen Y, Zheng Z, Wei X, Huang B, Shan Z, Liu J, Fan S, Chen J, Zhao F. Homocysteine induces oxidative stress and ferroptosis of nucleus pulposus via enhancing methylation of GPX4. Free Radic Biol Med. 2020 Sep 5;160:552-565.
- Kowluru RA. Diabetic Retinopathy: Mitochondria Caught in a Muddle of Homocysteine. J Clin Med. 2020 Sep 19;9(9):E3019.
- Aghayan SS, Farajzadeh A, Bagheri-Hosseinabadi Z, Fadaei H, Yarmohammadi M, Jafarisani M. Elevated homocysteine, as a biomarker of cardiac injury, in panic disorder patients due to oxidative stress. Brain Behav. 2020 Sep 23:e01851.
- CalikoÄŸlu M, Unlü A, Tamer L, Ercan B, BuÄŸdayci R, Atik U. The levels of serum vitamin C, malonyldialdehyde and erythrocyte reduced glutathione in chronic obstructive pulmonary disease and in healthy smokers. Clin Chem Lab Med. 2002 Oct;40(10):1028-31
- Agacdiken A, Basyigit I, Ozden M, Yildiz F, Ural D, Maral H, Boyaci H, Ilgazli A, Komsuoglu B. The effects of antioxidants on exercise-induced lipid peroxidation in patients with COPD. Respirology. 2004 Mar;9(1):38-42.
- Faucher M, Steinberg JG, Barbier D, Hug F, Jammes Y. Influence of chronic hypoxemia on peripheral muscle function and oxidative stress in humans. Clin Physiol Funct Imaging. 2004 Mar;24(2):75-84.
- Foschino Barbaro MP, Serviddio G, Resta O, Rollo T, Tamborra R, Elisiana Carpagnano G, Vendemiale G, Altomare E. Oxygen therapy at low flow causes oxidative stress in chronic obstructive pulmonary disease: Prevention by N-acetyl cysteine. Free Radic Res. 2005 Oct;39(10):1111-8.
- Nadeem A, Raj HG, Chhabra SK. Increased oxidative stress and altered levels of antioxidants in chronic obstructive pulmonary disease. Inflammation. 2005 Feb;29(1):23-32.
- Parija M, Bobby Z, Kumar V, Saradha B. Oxidative stress and protein glycation in patients with chronic obstructive pulmonary disease. Indian J Physiol Pharmacol. 2005 Jan;49(1):95-8.
- Rai RR, Phadke MS. Plasma oxidant-antioxidant status in different respiratory disorders. Indian J Clin Biochem. 2006 Sep;21(2):161-4.
- Van Helvoort HA, Heijdra YF, Thijs HM, Viña J, Wanten GJ, Dekhuijzen PN. Exercise-induced systemic effects in muscle-wasted patients with COPD. Med Sci Sports Exerc. 2006 Sep;38(9):1543-52.
- Jammes Y, Steinberg JG, Ba A, Delliaux S, Brégeon F. Enhanced exercise-induced plasma cytokine response and oxidative stress in COPD patients depend on blood oxygenation. Clin Physiol Funct Imaging. 2008 May;28(3):182-8.
- Mercken EM, Gosker HR, Rutten EP, Wouters EF, Bast A, Hageman GJ, Schols AM. Systemic and pulmonary oxidative stress after single-leg exercise in COPD. Chest. 2009 Nov;136(5):1291-1300.
- Tsai JJ, Liao EC, Hsu JY, Lee WJ, Lai YK. The differences of eosinophil- and neutrophil-related inflammation in elderly allergic and non-allergic chronic obstructive pulmonary disease. J Asthma. 2010 Nov;47(9):1040-4.
- Ahmad A, Shameem M, Husain Q. Altered oxidant-antioxidant levels in the disease prognosis of chronic obstructive pulmonary disease. Int J Tuberc Lung Dis. 2013 Aug;17(8):1104-9.
- Arja C, Surapaneni KM, Raya P, Adimoolam C, Balisetty B, Kanala KR. Oxidative stress and antioxidant enzyme activity in South Indian male smokers with chronic obstructive pulmonary disease. Respirology. 2013 Oct;18(7):1069-75.
- Ben Moussa S, Sfaxi I, Tabka Z, Ben Saad H, Rouatbi S. Oxidative stress and lung function profiles of male smokers free from COPD compared to those with COPD: a case-control study. Libyan J Med. 2014 Jun 12;9:23873.
- Elmasry SA, Al-Azzawi MA Ghoneim AH Nasr MY AboZaid MMN. Role of oxidant–antioxidant imbalance in the pathogenesis of chronic obstructive pulmonary disease. Egypt J Chest Dis Tuberc. 2015;64(4): 813-820.
- Ismail M, Hossain MF, Tanu AR, Shekhar HU. Effect of spirulina intervention on oxidative stress, antioxidant status, and lipid profile in chronic obstructive pulmonary disease patients. Biomed Res Int. 2015;2015:486120.
- Kodama Y, Kishimoto Y, Muramatsu Y, Tatebe J, Yamamoto Y, Hirota N, Itoigawa Y, Atsuta R, Koike K, Sato T, Aizawa K, Takahashi K, Morita T, Homma S, Seyama K, Ishigami A. Antioxidant nutrients in plasma of Japanese patients with chronic obstructive pulmonary disease, asthma-COPD overlap syndrome and bronchial asthma. Clin Respir J. 2017 Nov;11(6):915-924.
- Al-Azzawi MA, Ghoneim AH, Elmadbouh I. Evaluation of Vitamin D, Vitamin D Binding Protein Gene Polymorphism with Oxidant - Antioxidant Profiles in Chronic Obstructive Pulmonary Disease. J Med Biochem. 2017 Oct 28;36(4):331-340.
- Jha S, Bhattacharjee D, Chowdhuri S, Mitra A, Das A, Mukherjee K. Oxidants and Antioxidants in COPD Associated with Tobacco Smoke and Biomass Exposure. J. Evolution Med. Dent. Sci. 2019;8(46):3449-3453.
- Saribal D, Hocaoglu-Emre FS, Aydemir B, Akyolcu M. Determination of Relationship Between Lipid Peroxidation, Antioxidant Defence, Trace Elements and Hemorheology in COPD. Trace Elem Electroly. 2019;36(3):131-136
- Fläring UB, Rooyackers OE, Hebert C, Bratel T, Hammarqvist F, Wernerman J. Temporal changes in whole-blood and plasma glutathione in ICU patients with multiple organ failure. Intensive Care Med. 2005 Aug;31(8):1072-8.
- Biljak VR, Rumora L, Cepelak I, Pancirov D, Popović-Grle S, Sorić J, Grubisić TZ. Glutathione cycle in stable chronic obstructive pulmonary disease. Cell Biochem Funct. 2010 Aug;28(6):448-53.
Reviewer 2 Report
The authors demonstrated the antioxidant effects of pirfenidone and nintedanib, which has therapeutic effects for pulmonary fibrosis, in patients with that disease. The findings are quite important in clinical data in actual patients. However, there are several points which should be clarified for publication in this journal.
(1) Table 2 indicated that, after 24-week treatment, two agents significantly decreased oxidative stress in pulmonary fibrosis patients. Nevertheless, it seems that lung functions deteriorated. It is quite strange. Are there any advantage in antioxidant effects in treatment of pulmonary fibrosis? The relation of antioxidant effects and treatment outcome should be investigated and shown.
(2) Tables 3 and 4 indicated that antioxidant effects of two agents were not identical. Antioxidant effects and treatment outcome should be compared between two agents.
(3) d-ROMS, which is referred to as a crude marker of LOOH, may well-reflect total oxidative stress status in human more than other specific markers (Medicine (Baltimore). 2018 Nov;97(47):e12845. doi: 10.1097/MD.0000000000012845). For understanding overall oxidative stress and inflammation status in pulmonary fibrosis patients, this analysis should be performed and presented.
Author Response
The authors demonstrated the antioxidant effects of pirfenidone and nintedanib, which has therapeutic effects for pulmonary fibrosis, in patients with that disease. The findings are quite important in clinical data in actual patients. However, there are several points which should be clarified for publication in this journal.
(1) Table 2 indicated that, after 24-week treatment, two agents significantly decreased oxidative stress in pulmonary fibrosis patients. Nevertheless, it seems that lung functions deteriorated. It is quite strange. Are there any advantage in antioxidant effects in treatment of pulmonary fibrosis? The relation of antioxidant effects and treatment outcome should be investigated and shown.
We wish to thank the Reviewer for highlighting this point. Although counterintuitive, lung function deterioration during treatment is common. As described in the introduction “In 2014, the Food and Drug Administration approved nintedanib and pirfenidone for IPF treatment because of their ability to retard disease progression”. Thus, these drugs do not improve lung function but, rather, retard disease progression. The FVC deterioration observed in our study, 96mL with pirfenidone and 92mL with nintedanib, is similar to that previously reported in clinical trials [1-2]. The relationship between antioxidant effect and treatment outcomes should be properly investigated in larger placebo controlled studies. For these reason, our final sentence states “All reported data, including non-significant differences, may assist with the adequate design (i.e. sample size and/or biomarker selection) of larger interventional trials to confirm these findings and to establish whether such effects are associated with clinical improvement in this group”.
(2) Tables 3 and 4 indicated that antioxidant effects of two agents were not identical. Antioxidant effects and treatment outcome should be compared between two agents.
Please see response for question 1.
(3) d-ROMS, which is referred to as a crude marker of LOOH, may well-reflect total oxidative stress status in human more than other specific markers (Medicine (Baltimore). 2018 Nov;97(47):e12845. doi: 10.1097/MD.0000000000012845). For understanding overall oxidative stress and inflammation status in pulmonary fibrosis patients, this analysis should be performed and presented.
We thank the Reviewer for this suggestion. However, the use of d-ROMS to directly detect reactive oxygen species (ROS) is marred by several issues, particularly the ROS instability in biological samples [3]. Therefore, their quantification is inevitably affected by the time interval between sample collection and the assay, with longer delays associated with greater loss of ROS. Another concern is the possible interference from ceruloplasmin (a copper-containing enzyme responsible for oxidation of serum iron and aromatic amines and phenols) activity, which creates false positive signals [3-5]. Furthermore, iron, albumin, and thiol levels can also alter the test’s outcome [3]. For these and other reasons [6-8], we used other assays that are able to detect stable products of ROS.
- King TE Jr, Bradford WZ, Castro-Bernardini S, Fagan EA, Glaspole I, Glassberg MK, Gorina E, Hopkins PM, Kardatzke D, Lancaster L, Lederer DJ, Nathan SD, Pereira CA, Sahn SA, Sussman R, Swigris JJ, Noble PW; ASCEND Study Group. A phase 3 trial of pirfenidone in patients with idiopathic pulmonary fibrosis. N Engl J Med. 2014;370:2083-92.
- Richeldi L, du Bois RM, Raghu G, Azuma A, Brown KK, Costabel U, Cottin V, Flaherty KR, Hansell DM, Inoue Y, Kim DS, Kolb M, Nicholson AG, Noble PW, Selman M, Taniguchi H, Brun M, Le Maulf F, Girard M, Stowasser S, Schlenker-Herceg R, Disse B, Collard HR; INPULSIS Trial Investigators. Efficacy and safety of nintedanib in idiopathic pulmonary fibrosis. N Engl J Med. 2014;370:2071-2082.
- Kilk K, Meitern R, Härmson O, Soomets U, Hõrak P. Assessment of oxidative stress in serum by d-ROMs test. Free Radic Res. 2014 Aug;48(8):883-9.
- Harma MI, Harma M, Erel O. d-ROMs test detects ceruloplasmin, not oxidative stress. Chest 2006;130:1276; author reply 1276–1277.
- Gutteridge JM. Caeruloplasmin: a plasma protein, enzyme, and antioxidant. Ann Clin Biochem 1978;15:293–296
- Harma MI, Harma M, Erel O. Are d-ROMs and FRAP tests suitable assays for detecting the oxidative status? Eur J Obstet Gynecol Reprod Biol 2006; 127: 271–275.
- Buico A, Cassino C, Ravera M, Betta PG, Osella D. Oxidative stress and total antioxidant capacity in human plasma. Redox Rep. 2009;14(3):125-31.
- Harma MI, Harma M, Erel O. The FORT test: a novel oxidative stress marker or a well-known measure of ceruloplasmin oxidase activity? Atherosclerosis 2006; 187: 441–442
Round 2
Reviewer 1 Report
The authors addressed all my comments. All the responses are clear and, I appreciated the attempt to explain the missing points and the shortcomings of the study. Nevertheless, in my opinion, the fact that they did not measure any of the most reliable markers of oxidative stress, remains an intrinsic and unsolved issue of the work. How can we know whether the two drugs really work against oxidative stress, if we do not evaluate this condition properly? This methodological problem is relevant especially considering the high impact and important ranking of the journal in the redox-biology field
Author Response
While we respect the Reviewer’s comments, we remain of the opinion that the oxidative stress biomarkers measured in our study have been widely investigated in previous studies. Such studies, described in our initial rebuttal, have described not only their pathophysiological role but also their clinical significance. In particular, as also previously highlighted, over the last decade our group has published a considerable number of studies on GSH and MDA as robust biomarkers of oxidative stress.
Reviewer 2 Report
I have no further criticism for this revised manuscript.
Author Response
Thank you